# Assessment of the Potential of Waste Copper Chromium and Arsenic (CCA)-Treated Timber Fibre Reinforced Polypropylene Composites for Construction

Jacob Nelson, Kim L. Pickering and Mohammad D. H. Beg *

School of Engineering, University of Waikato, Private Bag-3105, Hamilton 3216, New Zealand
* Correspondence: mdhbeg@waikato.ac.nz

**Abstract:** This paper investigates the potential of recycling waste copper chromium and arsenic (CCA)-treated timber for use as a reinforcement material in wood–plastic composites (WPCs) produced for use in construction, including an assessment of mechanical properties and the leaching of heavy metals. Wood flour was obtained through mechanical grinding, and fibres were obtained through alkaline digestion followed by bleaching. Composites produced with 40 wt.% bleached fibres showed increased tensile strength from 18.5 MPa for the polypropylene used as the matrix to 27.6 MPa. Likewise, the Young's modulus was increased from 0.84 to 2.33 GPa. The treatment of fibres was found to reduce arsenic concentration by up to 99.9%. Furthermore, the arsenic in the leachate from composites was found to decrease from 41.29 to 0.07 ppb when comparing CCA-treated wood flour composites to bleached fibre composites. The composites' material properties indicate that the use of end-of-life CCA-treated timber could be used to produce a composite material that could be used in New Zealand's building sector to meet the requirements of semi-structural applications.

**Keywords:** CCA-treated timber; wood plastic composites; creep; leaching

## 1. Introduction

The construction sector is expected to contribute 40–50% of the waste produced in New Zealand [1]. Transitioning to a circular economy could see the use of waste materials from the construction sector as raw materials for new construction. This would help reduce reliance on virgin materials, eliminate waste streams, and reduce emissions emanating from the extraction and processing of new materials for construction [2].

Copper, chromium, and arsenic (CCA)-treated timbers are used extensively in the construction sector in New Zealand due to its high resistance to insects, fungi and water damage caused by the addition of preservatives [3]. In 2006, it was reported that CCA-treated timber was the most-produced treated timber in New Zealand, making up an estimated 69% (by volume) of the treated timber in New Zealand [4]. However, in the current state of the construction sector in New Zealand, CCA-treated timber is typically sent to landfill due to issues with leaching in clean fill, and the potential for emissions of toxic vapours to the atmosphere during incineration [5,6]. The use of CCA-treated timber as a raw material for wood–plastic composite (WPC) would allow for a product to be created directly from waste material from construction, to be used back in the construction sector. Furthermore, WPC has shown the potential to be reprocessed with minimal loss in mechanical properties [7]. This indicates that it could be recycled and kept within the industry for long periods of time and providing the potential to shift the current waste of CCA-treated timber to being reused circularly.

Previous studies on waste CCA-treated timber as a reinforcement material for WPC have found limitations, primarily due to the environmental and health concerns of the heavy metals within the timber and corresponding composites [8–10]. However, these studies have primarily focused on using wood flour. This necessitated the use of chemical methods,

rather than mechanical processing, to reduce heavy metal content and possibly obtain better reinforcement that would allow the material to be used in high value applications.

Earlier work has supported the notion that the mechanical performance of WPC produced with CCA-treated timber can be better than that of WPC produced with virgin pine. Furthermore, heavy metals were found to increase the hydrophobicity of the timber and increase thermal dispersion during processing. This increases the flow of the thermoplastic, thereby providing better encapsulation, which results in improved interfacial strength and mechanical performance. Resistance to biological decay was also found to increase when compared to using virgin timber as a reinforcement material in WPC. However, leaching of arsenic from the WPC was found to be substantially higher than the maximum allowable value (MAV) of 10 ppb in drinking water [9,10]. Kamdem et al. found that the leaching became stable after 400 h and was negligible after 1000 h, and the concentration of As in the leachate reached 12 ppm. The use of polypropylene graft maleic anhydride (MAPP) as a coupling agent in composites was found to reduce the concentration of As leached from 1.5 to 0.5 ppm; however, leachate levels were still much higher than the MAV in drinking water [10].

More recently, researchers have investigated the reinforcement of polypropylene with fibres obtained from CCA-treated timber through alkali chemical treatments. Their study found that a tensile strength of 29 MPa and a Young's modulus of 2.5 GPa were attainable with 30 wt% fibre. The research suggested that the fibres obtained from the treated timber resulted in less moisture absorption of corresponding composites than virgin timber fibre composites [8]. However, the study did not investigate the effects of the chemical treatment on the fibres' heavy metal content or the composites' leaching potential.

Alkali processes are commonly used in the pulp and paper industry to extract fibres from timber. Alkali treatments are also commonly performed to remove lignin, hemicellulose, and extractives and increase surface area and roughness to improve interfacial strength. For natural fibre composites, the use of alkali treatments has been widely reported to improve interfacial strength, tensile strength, Young's modulus, failure strain, impact strength, fracture toughness, and flexural properties of natural fibre composites [11]. Literature on the alkaline treatment of CCA-treated timber for use in composites is minimal, and optimising the process would likely further improve the mechanical performance of composites [8]. The literature also failed to evaluate the effects of treatment on the heavy metals within the timber, and the possibility of leaching from the composites.

Hydrogen peroxide bleaching is a commonly used step in the bleaching of pulp and is considered an environmentally friendly solution, as it reduces the number of chlorine-based chemical bleaching processes used [12]. Hydrogen peroxide bleaching of fibre has also been shown to improve composite performance. A recent study found an increase in kenaf fibre and Poly(lactic acid) (PLA) composite tensile strength from 38.5 to 42.9 MPa (with 10 wt% fibre) when the fibres were treated using hydrogen peroxide. The study suggested that increased properties were likely due to increased fibre roughness, resulting in increased interfacial strength within the composites [13]. Interestingly, the use of hydrogen peroxide as a pulping chemical for CCA-treated timber has also been investigated, and it has been found that the use of hydrogen peroxide in a pulping process could remove 100% Cr and As and 95% of Cu in some cases [14].

The environmental impact of fibre treatments is of concern due to the high chemical and energy consumption of common processes. Ultrasonic treatment of lignocellulosic biomass has shown the potential to assist with commonly used mechanical and chemical treatment processes, reducing the time of treatments and chemical consumption, therefore potentially reducing the environmental impact [15]. However, the treatment does consume a lot of energy [15]. In simple terms, ultrasonic treatment creates large localised pressure differences within a solution to enhance mechanical or chemical treatment processes [15]. Ultrasonic treatment of wood flour has been investigated as a treatment method for wood–plastic composites, and authors reported that the use of ultrasonic treatment increased lignin removal from the wood flour and increased the tensile strength and modulus of

composites. The study indicated that the improvement in properties was due to the reduction of lignin and increased exposure of cellulose, which improves its bonding ability with an MAPP coupling agent. The report also indicated some reduction in tensile strength in composites treated with higher concentrations of NaOH, most likely due to the degradation of cellulose [16]. Ultrasonic treatment also has the potential to induce fibrillation on natural fibre surfaces [17]. The potential to generate fibrillation of the fibre surface can increase the surface area of fibres and therefore increase interfacial strength within composites. This phenomenon has been shown using fibre beating to induce fibrillation. Fibre beating was found to cause fibrillation of the primary cell wall, increasing surface area, and therefore increasing interfacial strength. The study found an increase of up to 10% in tensile strength; although, with increased fibre beating, the tensile strength was reduced, likely due to a reduction in fibre length [18].

Generally, there have been studies on alkali digestion, bleaching or ultrasonication for some common fibres. However, there are no studies on the combination of alkaline, bleaching, and ultrasonication and the removal efficiency on Cu, Cr and As from recycled CCA-treated timber. Therefore, the aim of this study was to investigate the use of alkali digestion, hydrogen peroxide bleaching, and ultrasonic treatment to obtain fibre from waste CCA-treated timber and wood flour in wood–plastic composite materials to assess the potential of such recycled materials in the construction sector.

## 2. Materials and Methods

### 2.1. Materials

CCA-treated timber was supplied as Radiata pine treated to an H 3.2 hazard class, supplied as approximately 1-metre offcuts of $2 \times 4''$ and $2 \times 2''$ planks by ITM, Hamilton, New Zealand. Yuplene RX 3600 Polypropylene (PP) Random Copolymer produced by SK Global Chemical, Seoul, South Korea, was used as the matrix. AC 950P Propylene Maleic Anhydride Copolymer (MAPP) produced by Honeywell International Inc., Charlotte, NC, USA, was used as a coupling agent. Bulk grade solid NaOH was used for the alkali digestion. Hydrogen peroxide (30%) was obtained from Perhydrol™, and sodium silicate solution obtained from Sigma-Aldrich, St. Louis, MO, USA, was used in bleaching Emsure. Nitric acid (65%) and Emsure Hydrochloric acid fuming (37%) were used for preparing samples for ICP–MS analysis.

### 2.2. Methods

2.2.1. Production of Reinforcement Materials

The timber was initially chipped using a Ryobi RSH 2455 G Garden Shredder and then granulated with a Moretto GR granulator with an 8 mm grate. It was then dried for at least 12 h in an 80 °C oven before further processing. Wood flour was produced via grinding granules of timber using a Sunbeam Multigrinder to make fine wood particles. About 10 g batches of dried timber granules were ground for 1 min periods and then screened through an 850 μm sieve.

Digestion of timber to obtain fibres was performed using 10 wt% NaOH aqueous solution in a custom laboratory-scale batch digester. The system was heated from 25 °C (room temperature) to 160 °C over 2 h and then held at 160 °C for 2 h. Digestion took place with a 1:8 ratio of timber to solution (by weight). The system was then quenched to approximately 50 °C, and the fibres were washed until a pH of 7 was attained.

Bleaching of the digested fibres was undertaken using a solution of hydrogen peroxide and sodium silicate. About 25 g of oven-dry fibres and 2 L of tap water were added and mixed in a beaker creating a 1:80 ratio (by weight). A Silverson L4RT homogeniser was used to mix the fibre into the water and ensure the fibres were saturated. The mixture was then heated on a heating plate until a temperature of 70 °C was reached. In total, 100 mL of hydrogen peroxide and 50 mL of sodium silicate were then added to start the bleaching reaction. The bleaching process was undertaken for 10 min on the heating plate, while stirring continuously. The fibres were washed until a pH of 7 was attained.

Before being used in fibre testing or for compounding, all reinforcement was dried at 80 °C in an oven for at least 12 h. The fibres were also prone to 'clumping' together upon drying and required separating before composite production. A Sunbeam Multigrinder was used to shear the fibres apart with blunt blades at a high rotational speed. Fibres were separated using 3 g of fibre for 10 s periods. Any clumps of fibre that failed to separate was subjected to repeated processing, as many times as required, until separation occurred.

### 2.2.2. Analysis of Fibre

Scanning electron microscopy (SEM) was used to visually investigate the fibre surfaces. Fibres were mounted on aluminium stubs with carbon tape before being coated with a thin layer of platinum to make them conductive using a Hitachi E-1030 Ion Sputter. The fibres were examined using a Hitachi S 4700 Field Emission Gun Scanning Electron Microscope operating at 5 kV.

Fibre quality analysis was undertaken using a Fibre Quality Analyser—360 (FQA-360). The analysis was performed to provide data on fibre length, width, aspect ratio, number of bundles, and percentage of fines present. Fibre length was measured following ISO 16065-1: 2014 Determination of fibre length by automated optical analysis [19].

The thermal properties of fibres and wood flour were assessed using a Netzsch$^{®}$ STA449 F5 Jupiter thermogravimetric analyser (TGA). A dynamic analysis was carried out from 30 to 600 °C at a heating rate of 10 °C/min under argon flow of 40 mL/min.

Inductively Coupled Plasma–Mass Spectrometry (ICP–MS) analysis was performed to measure the heavy metal content in wood flour and fibres to assess the changes in copper, chromium, and arsenic content, after chemical and mechanical treatments. Wood fibre and flour samples were prepared for ICP–MS analysis following a full sediment digestion method. A blank sample was also produced as a reference for the analysis. Roughly 0.2 g of the sample was weighed, recorded to 4 d.p., and placed in a clean 50 mL falcon tube. A total of 1 mL of $HNO_3$ and 0.33 mL of HCl were added to the falcon tube. Type 1 water was added to ensure submersion of the samples and left to predigest in a fume hood overnight. A digestion block was then heated to 80 °C, and the samples were heated for 1 h. Type 1 water was added to each falcon tube to top up the sample to 50 mL. The samples were centrifuged for 10 min at 3000 rpm and the supernatant was filtered using a 0.45 μm cellulose acetate, Minisart Syringe filter. In total, 10 mL of the sample was then added to a 15 mL falcon tube for ICP–MS analysis. The analysis was performed using an Agilent 8900 (Agilent Technologies, Santa Clara, CA, USA) controlled by MassHunter Workstation (version 4.5). Triplicates of each sample were assessed to ensure accuracy in the measurement.

### 2.2.3. Production of Composites

Composites were compounded using a Labtech LTE-20-44 twin-screw extruder. The extruder was purged before extrusion and between different samples of composites. Extrusion was performed using a screw speed of 40 rpm and barrel temperatures ranging from 150 to 170 °C. After compounding, the composite was granulated using a Moretto GR granulator with an 8 mm grate.

The WPC granules were injection moulded using a BOY 35A Injection moulding machine and a mould for a standardised Type 1 (ASTM 638 [20]) Dog-Bone, and a beam for 3-point testing (ISO 178 [21]).

### 2.2.4. Characterisation of Composites

Tensile testing was carried out based on ASTM 638-03: Standard Test Method for Tensile Properties of Plastics [20]. Test specimens were placed in a Binder GmbH Model KMF 115 conditioning chamber at 23 ± 3 °C and 50 ± 5% relative humidity for at least 48 h before testing was undertaken. The specimens were then tested using an Instron 5982 Tensile Testing machine, using a 5 kN load cell operated at 5 mm/min. An Instron

2630-107 extensometer was used to measure the strain. Five specimens were tested per condition for composites.

A basic creep test was undertaken to investigate the time-dependent mechanical properties of composites and polypropylene. Test specimens were placed in a conditioning chamber at $23 \pm 3$ °C and $50 \pm 5\%$ relative humidity for at least 40 h. The specimens were tested in 3-point bending using an Instron 5982 Tensile Testing machine, using a 5 kN load cell. Creep properties were tested for polypropylene and composites at 15 and 30% of the maximum flexural strength (Measured based on ASTM D790-17: Standard Test for Flexural Properties of Unreinforced and Reinforced Plastics and Electrical Insulating Materials [22] using the ISO 178 [21] beams) of the corresponding material. The test was undertaken in two steps; the specimen was initially loaded to the desired stress at a rate of 20 MPa/min and then held at the set stress for 12 h.

Scanning electron microscopy (SEM) was used to observe the composite's fracture surfaces. Composites that had undergone tensile testing were mounted on aluminum stubs with carbon tape before being coated with a thin layer of platinum to make them conductive using a Hitachi E-1030 Ion Sputter. The fractured surfaces were examined using a Hitachi S 4700 Field Emission Gun Scanning Electron Microscope operating at 5 kV.

Leaching analysis was completed based on AWPA E11-16 Standard Method for Accelerated Evaluation of Preservative Leaching [23]. The leaching of heavy metals was completed by submerging three material beams (approximately $80 \times 10 \times 4$ mm) in 300 mL of distilled water. Leachate was collected, and the water was replaced at set time intervals including 6, 12, 24, 48 h and thereafter at 48 h time intervals until 14 days was reached. The leachate samples were then acidified by adding 0.2 mL of nitric acid to 9.8 mL of leachate and measured for heavy metal content using ICP–MS with the methods described in the ICP–MS analysis section above.

### 2.2.5. Statistical Analysis

A statistical analysis was performed for some of the testing where a number of samples were tested. The averages of each test are displayed for the material properties. The coefficient of variance (COV) was calculated as the standard deviation divided by the average value and displayed as a percentage. Differences in fibre and composite results were tested using a statistical analysis performed using a Tukey test and analysis of variance (ANOVA) for significant ($p < 0.05$) differences. All analyses were performed using MINITAB Statistical Software.

### 3. Results and Discussion

### *3.1. Characterisation of Fibre*

The wood fibre and flour samples used throughout the analysis were given coded names and acronyms, such as wood flour (F), digested fibre (D), digested and ultrasonic-treated fibre (DU), digested and bleached fibre (DB), and digested bleached and ultrasonic-treated fibre (DBU).

SEM was used to investigate the fibre surfaces after all processing steps and variables. The images of digested fibre show that the separation of fibres from the wood chips has started to occur (Figure 1A), and some single fibres are present. However, fibre bundles are still present, and further processing could improve separation. The texture of the fibre surface (Figure 1B) suggests that localised lignin and extractives remain on the surface [24]. The presence of fibre bundles further suggests lignin and extractives are present on the fibre surfaces, acting to bind the cellulosic fibres together in fibre bundles [7].

Figure 1C,D shows the surface of the digested and ultrasonic-treated fibres. The surfaces are visually smoother than those of the alkali pulped fibres, signifying that ultrasonic treatment has removed some of the lignin and extractives from the surface of the fibres. The images indicate that further fibre separation has occurred during the ultrasonic treatment, and fibrillation is starting to occur as small fibrils can be seen on the surface. Both fibres and cellulose fibrils within the fibres are held in place by lignin and extractives. Therefore,

the potential fibre separation and fibrillation of the fibre surface indicate further removal of lignin and extractives holding the fibres and microfibrils together [7]. The potential reduction of lignin and extractives agrees with the literature, where ultrasonic treatment has been found to remove lignin and extractives from wood flour [16].

Figure 1E,F shows the fibre surface of the digested and bleached fibres. The fibres in the images are seen to have a more textured surface than the digested fibre. This surface is likely a result of the removal of localised lignin and extractives from the fibre surfaces during the bleaching process, exposing the microfibrils in the fibre. This theory agrees with the literature, where a rougher surface was obtained from hydrogen peroxide bleaching, where it was subsequently used to improve interfacial strength in the corresponding composites [13]. Hydrogen peroxide bleaching is also known to remove lignin content [25]. The images also imply that more fibre separation has occurred, and more individual fibres are present (compared to the digested fibre). The potential increase in fibre separation is likely the result of the removal of lignin and extractives, which naturally hold the fibres together in fibre bundles.

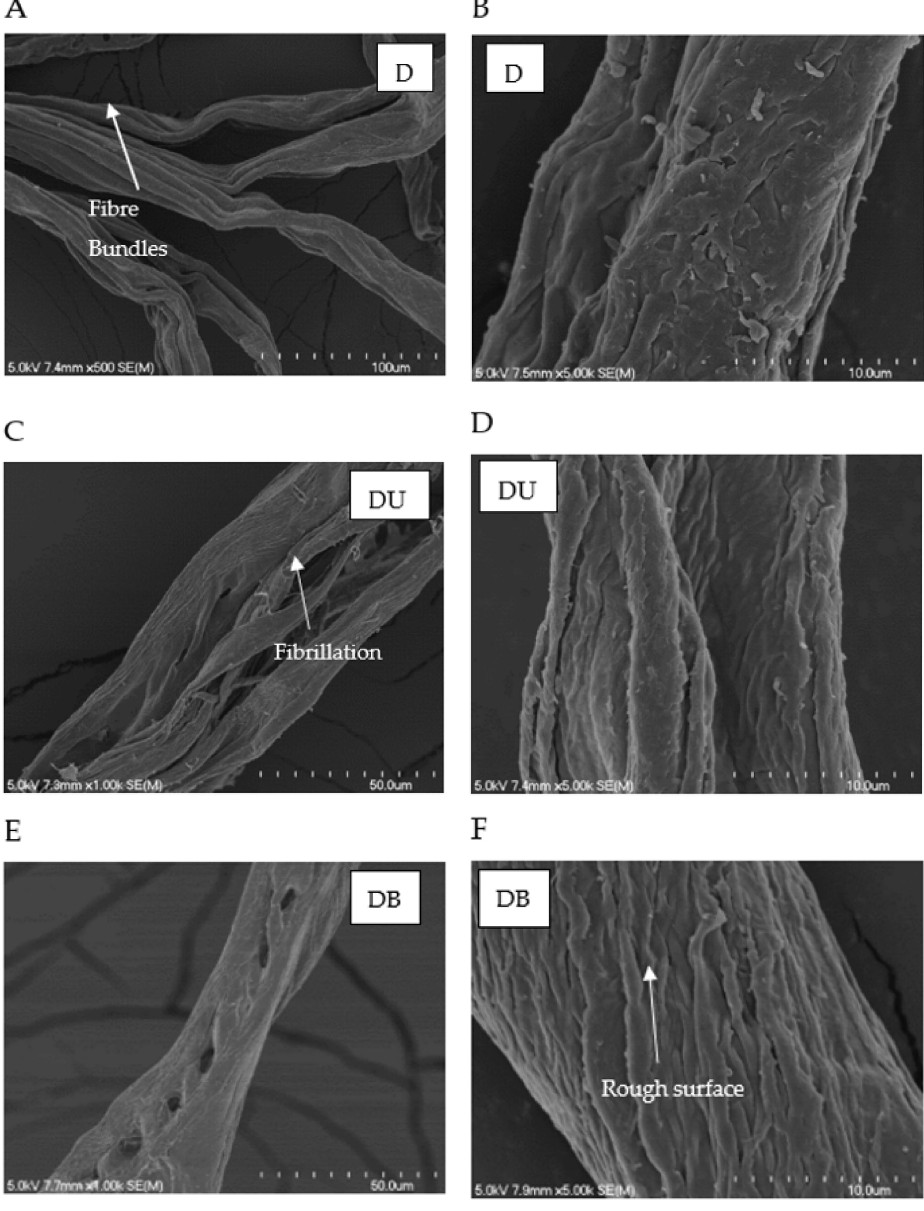

**Figure 1.** *Cont.*

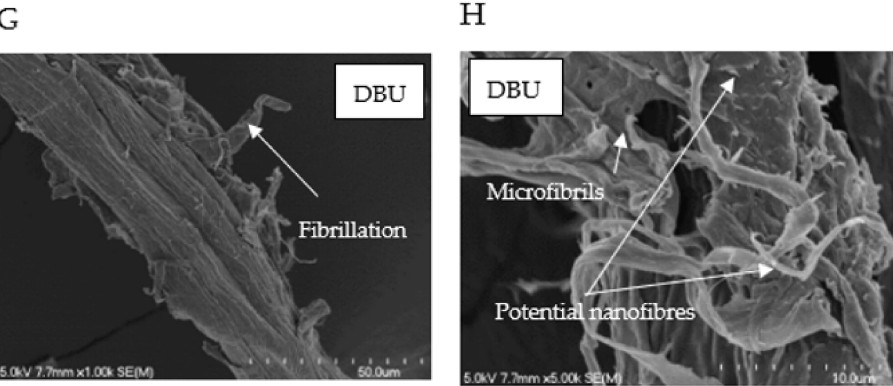

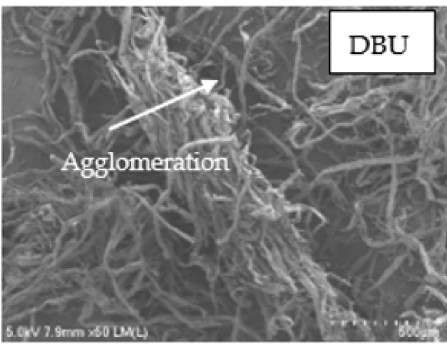

**Figure 1.** SEM of (**A,B**)—digested fibre (D), (**C,D**)—digested and ultrasonic-treated fibre (DU), (**E,F**)—digested and bleached fibre (DB), and (**G–I**)—digested bleached and ultrasonic-treated fibre (DBU).

The digested, bleached, and ultrasonic-treated fibres can be seen in Figure 1G–I. The use of ultrasonic treatment after bleaching the fibres shows significant fibrillation of the fibres and the exposure of microfibres and potentially nano fibres. The added fibrillation from the ultrasonic treatment increases the surface area of the fibres, suggesting a potential for more interaction and better interfacial strength between the fibres and matrix.

The fibrillation induced by the ultrasonic treatment can increase the interfacial strength of composites through the presence of microfibrils and increased surface area. However, literature has equally shown that mechanical processing to induce fibrillation can also damage the fibres and reduce fibre length, with the potential of decreasing composite strength [7]. Figure 1G,H shows significant fibrillation induced by the ultrasonic treatment after the bleaching process, suggesting potential for the fibre strength and length to be decreased due to the ultrasonic process, and therefore potential to reduce corresponding composite mechanical properties.

Figure 1I reveals that the fibres tend to agglomerate after the treatment, as shown by the large group of fibres. This phenomenon has the potential to reduce the mixing of matrix and fibres during compounding, therefore reducing the wetting of fibres and reducing composite properties.

The results from the fibre quality analysis were used to study the fibres' geometry, including width, length, aspect ratios, number of shives (bundles), and fines. Table 1 summarises the information recorded for all fibre processing variables.

It can be seen that bleaching of the digested fibres reduced the diameter of the fibres without any effect on fibre length. Literature suggests that increased chemical processing results in a reduction in fibre width and the reduction in corresponding kappa number, indicating the removal of lignin [26]. Therefore, the reduction in fibre width due to the bleaching further suggests the removal of lignin and extractives from the fibre surface, reinforcing the results of the SEM. The number of bundles or shives is also seen to decrease

after the bleaching of fibres, which further suggests the reduction of lignin and extractives in the fibres, which are known to hold fibre bundles together [7].

**Table 1.** Summary of results from fibre quality analysis. The number of shives as a percentage of the number of fibres measured is given in parenthesis.

| Sample | | Number of Fibres Measured | Mean Length (mm) | Mean Width (μm) | Aspect Ratio | Number of Shives | Fines (Length Weighted) (%) |
|---|---|---|---|---|---|---|---|
| Digested | (D) | 33,190 | 1.20 | 32.39 | 36.95 | 85.67 | 1.01 |
| Digested + Ultrasonic | (DU) | 38,720 | 0.94 | 32.62 | 28.80 | 10.33 | 2.74 |
| Digested + Bleached + | DB | 47,840 | 1.18 | 31.47 | 37.64 | 19.00 | 1.03 |
| Digested + Bleached + Ultrasonic + | DBU | 36,430 | 0.66 | 32.76 | 20.10 | 66.00 | 16.66 |

Ultrasonic treatment reduced the mean fibre length and had minimal effect on the fibre width, resulting in a lower aspect ratio. However, the results suggest that ultrasonic treatment (Sample DU) reduced the number of shives or fibre bundles, reinforcing the results of the SEM where more individual fibres were visually present, and separation of fibre bundles had been improved.

The ultrasonic treatment of the digested and bleached fibres was found to reduce the length of the fibres from 1.18 to 0.66 mm, a reduction of 44%. Furthermore, the percentage of fines, weighted by length, was increased from 1.03% to 16.66%. Although the SEM suggested the potential to increase interfacial strength due to fibrillation of the fibres, the large reduction in fibre length and increased number of fines in the fibres further suggests the potential of reducing composite properties due to reduced fibre reinforcement capability.

The analysis shows that the use of ultrasonic treatment increases the fibre width. After ultrasonic treatment, the mean fibre width of the digested fibres slightly increased from 32.39 to 32.62 μm (comparing D and DU fibre) and increased the mean width of the bleached fibres from 31.47 to 32.76 um (comparing DB and DBU fibre). The apparent increase in width is likely due to the additional fibrillation surrounding the fibres, which resulted in an apparent fibre width that is larger than the actual fibres. The SEM reinforces this, as the difference was larger in the width of the bleached fibres, where substantially more fibrillation was indicated. It is likely that the actual width of the fibres has been reduced due to the microfibrils being separated from the primary wall of the cellulosic fibres and the potential removal of lignin from the fibre surface.

The results of the analysis also show that the number of shives increased with the ultrasonic treatment in the digested and bleached fibres (comparing DB and DBU fibre). The shives measured are likely a result of the agglomeration of fibres, indicated in Figure 1I of the SEM results, being measured as shives.

The digestion process aimed to reduce the content of lignin, hemicellulose, and extractives, in a single step. Literature on Kraft pulping indicates that hemicellulose is originally removed at a faster rate than lignin [27]. The digestion process also appears to follow this trend, as the spectra in Figure 2 indicates the removal of most of the hemicellulose in the digestion process, whereas residual lignin appears to remain. A clear peak in the wood flour spectra at 1726 cm$^{-1}$ is specific to -COOH and C=O functional groups, which are specifically found in hemicellulose. The spectra for all other fibres do not contain this peak, suggesting that hemicellulose has primarily been removed during the digestion process [28].

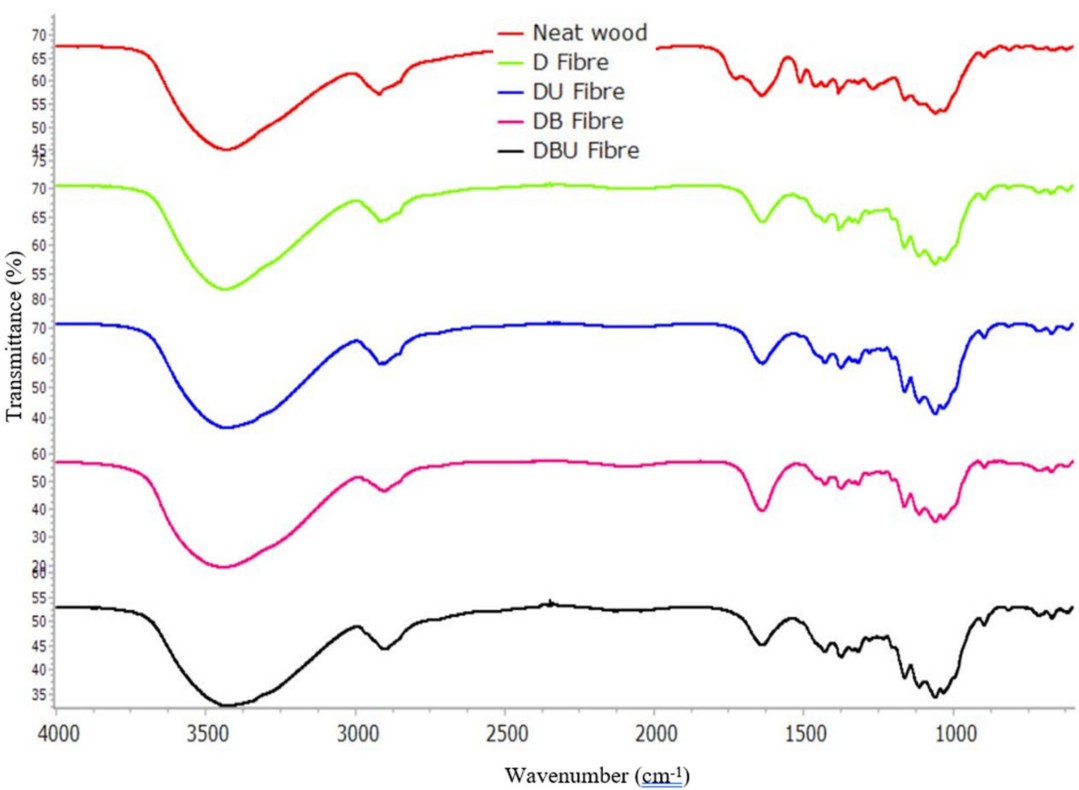

**Figure 2.** FTIR results from neat wood and all fibre samples. Neat wood (wood flour), D (digested fibre), DU (digested and ultrasonic-treated fibre), DB (digested and bleached fibre), and DBU (digested, bleached, and ultrasonic-treated fibre).

The spectra suggest a reduction of lignin in the digestion and bleaching processes. The peaks at 1511 cm$^{-1}$ represent the functional group C=C, and are assigned to aromatic rings of lignin, specifically indicating guaiacyl (G) units. The indication of G units of lignin makes sense, as the lignin in soft woods such as *Pinus radiata* is known to primarily be composed of G units [29]. The spectra show a reduction in the 1511 cm$^{-1}$ in the digested fibre and a further reduction in the peak after bleaching. The reduction in the size of the peak indicates partial removal of lignin from the fibre during these chemical processes. This removal of lignin is further inferred by reducing peaks at 1460 cm$^{-1}$, indicating the reduction in C-H functional groups assigned to lignin. However, lignin was still clearly present in all samples, shown by the small peaks still present at 1511 cm$^{-1}$, 1460 cm$^{-1}$, and the peaks remaining at 1120 cm$^{-1}$ and 1039 cm$^{-1}$, which all indicate functional groups found in lignin. The presence of lignin even after bleaching is expected; literature from the pulp and paper industry suggests that the bleaching of pulp can take up to seven different chemical treatments due to the difficulty of removing lignin [25,28].

The figure suggests that the use of ultrasonic treatment with NaOH gives no significant change in the peak, indicating little change in the constituents present in the fibres. However, the FTIR spectra used is qualitative, and the processing could have small unseen effects on the constituents present in the fibres.

X-ray diffraction (XRD) was used to analyse the crystallinity of the timber and fibres. The crystallinity index of the wood flour and fibres gives further insight into the constituents in the timber, and the XRD spectra provides insight into any potential changes to the crystalline cellulose. The crystallinity index was calculated using the Segal method using the spectra in Figure 3. The resulting crystallinity indices (Xc) are displayed in Figure 3 along with the spectra. The results show that the crystallinity index increases with progressing chemical processing. The crystallinity index increased from 41.9% for the timber, to 67.2% for the digested fibres (comparing F and D samples). Furthermore, the crystallinity index increased to 71.5% because of the bleaching process (comparing D

and DB samples). The increase in the crystallinity index was likely due to the removal of amorphous material, including lignin, hemicellulose, and other extractives. This reduction in amorphous constituents is confirmed by the SEM, fibre quality analysis, and FTIR discussed earlier.

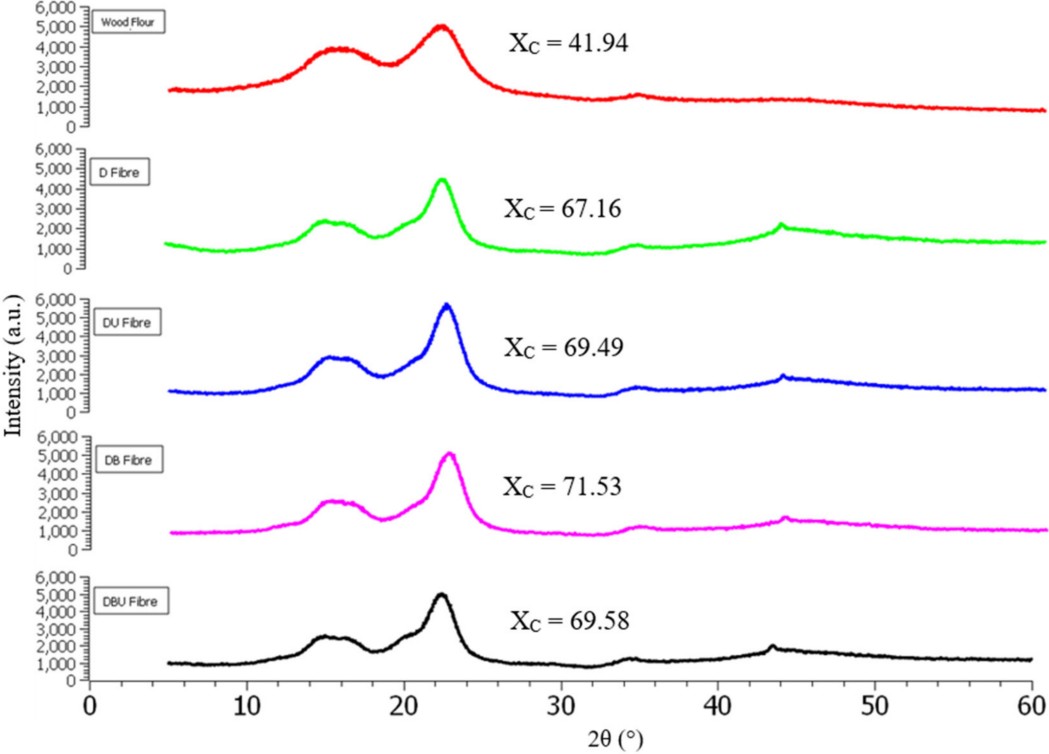

**Figure 3.** XRD of CCA results from treated timber and all fibre samples. F (wood flour), D (digested fibre), DU (digested and ultrasonic-treated fibre), DB (digested and bleached fibre), and DBU (digested, bleached, and ultrasonic-treated fibre).

The results from the XRD analysis show that the use of ultrasonic treatment after digestion also increases the crystallinity index, which is likely due to the removal of small amounts of amorphous constituents (comparing D and DU fibre). However, the crystallinity index is reduced after the ultrasonic treatment of the bleached fibres (comparing DB and DBU fibre). The spectra in Figure 3 show a shoulder peak emerging at roughly 20°, which is more obvious in the DBU fibre. The peak suggests the transformation from cellulose Iβ to cellulose II, resulting in the formation of the peak at 20° (corresponding to the 110 plane in cellulose II) and the reduction in the peak corresponding to the 002 plane. The potential transformation of small amounts of cellulose Iβ to cellulose II explains the reduction in the calculated crystallinity index caused by the high amounts of energy during the ultrasonic treatment. This reduction also shows the limitations of the Segal method used for calculating the crystallinity index [30,31].

TGA was used to analyse the thermal degradation of the treated timber and fibres. Figure 4 shows the TG and DTG (1st derivative of TG) analysis for all fibres and wood flour samples and Table 2 summarises the mass change, onset temperature, and peak degradation temperatures. The DTG results show the first derivative of the TG results, providing peaks that represent the rate of thermal degradation. Variances in the TG and DTG for the fibre samples are likely due to differences in the content of cellulose, lignin, hemicellulose, and extractives. Previous studies suggest that "hemicellulose decomposes at a lower temperature range (220–315 °C) than cellulose (300–400 °C), while lignin decomposes over a broad range of temperatures (150–900 °C)" [32]. However, it has been reported that the heavy metals can affect the thermal conductivity of the material, potentially affecting the thermal degradation properties [10].

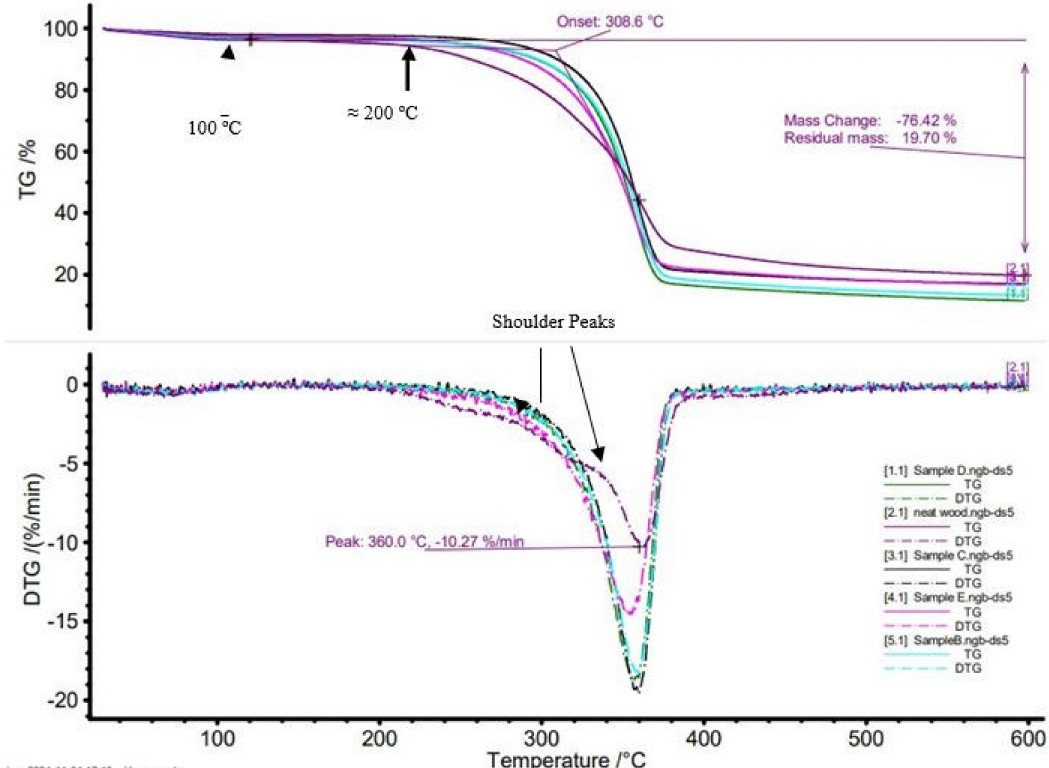

**Figure 4.** TGA results from treated timber and all fibre samples. Neat wood (wood flour), D (digested fibre), DU (digested and ultrasonic-treated fibre), DB (digested and bleached fibre), and DBU (digested, bleached and ultrasonic-treated fibre).

**Table 2.** Data from TGA analysis for treated timber and all fibre samples, and crystallinity index from XRD analysis.

| Sample | | Residual Mass (%) (Measured at 600 °C) | Onset Temperature (°C) | Peak Degradation Temperature (°C) |
|---|---|---|---|---|
| Treated timber | F | 19.7 | 308.6 | 360.0 |
| Digested | D | 13.3 | 327.6 | 359.7 |
| Digested and ultrasonic treated | DU | 17.0 | 330.9 | 359.6 |
| Digested and bleached | DB | 11.5 | 326.6 | 356.5 |
| Digested, bleached and ultrasonic treated | DBU | 16.7 | 316.1 | 353.9 |

The results of the TG testing indicate an initial change in mass at approximately 100 °C in all samples, suggesting the presence of water. The DTG of wood flour shows a major peak at 360 °C, and shoulder peaks at 260.7 and 326 °C, likely indicating the lignin and hemicellulose in the timber degrading before the cellulose. The samples all show initial thermal degradation at approximately 200 °C, most likely due to the presence of lignin, which degrades at lower temperatures than the other major constituents. This is most significant in the wood flour; however, it occurs in all samples.

ICP–MS was used to analyse the heavy metal content in the reinforcement materials. The results of the analysis are shown in Table 3. In New Zealand, H 3.2 CCA-treated timber requires the total concentration of heavy metals to be 0.37 wt.% (in the timber) as defined by NZS 3640 [33]. However, the sum of the three heavy metals in the timber adds to 0.42 wt.%, which is slightly higher than the New Zealand standard. The specific content of copper, chromium, and arsenic in the timber was measured to be 0.0839 wt.%, 0.13785 wt.%, and

0.0968 wt.%, respectively, inferring proportions of 26.8% Cu, 41.2% Cr and 32.0% As. The results fit reasonably well with the relative proportions required by NZS 3640 in the CCA treatment of 23–25% Cu, 38–45% Cr, and 30–37% As [33]. The copper concentration is the only heavy metal lying outside the expected range (1.3% higher than expected). The small differences in heavy metal content compared to the NZ standard are potentially due to variation within the timber and the small sample size of wood flour used for testing. However, the results are very similar to the New Zealand standard and can be regarded as an accurate representation of the trend in the heavy metal content in the wood flour and fibres.

**Table 3.** Mean heavy metal concentration in fibre and timber samples. F (wood flour), D (digested fibre), DU (digested and ultrasonic-treated fibre), DB (digested and bleached fibre), and DBU (digested, bleached and ultrasonic-treated fibre). The COV is presented in parenthesis (%).

|  | Cu (ppm) | Cr (ppm) | As (ppm) |
|---|---|---|---|
| F | 1126.4 (4.57) | 1728.5 (5.61) | 1346.0 (4.97) |
| D | 1093.6 (0.91) | 957.9 (0.71) | 1.94 (1.43) |
| DU | 986.89 (3.43) | 852.17 (5.42) | 1.30 (10.8) |
| DB | 556.53 (1.99) | 353.16 (2.52) | 1.21 (3.95) |
| DBU | 503.22 (4.95) | 338.76 (5.70) | 0.823 (9.09) |

Table 3 shows how the concentration of heavy metals changes with the treatment of the timber and fibres. The results show that the concentration of arsenic was reduced by 99.8% from the wood flour to the digested fibres and was further reduced with further treatment. The reduction in arsenic content signifies that the fibres could potentially be used safely in wood–plastic composite applications, as arsenic is considered the most toxic of the three heavy metals [34]. The concentration of copper and chromium was also reduced after digestion, bleaching, and ultrasonic treatments. The reduction in the concentration of chromium was less than that of arsenic (Cr was reduced from 1728.5 to 957.9 ppm from the treated timber to digested fibres, respectively). The concentration of copper was found to only decrease from 1126.4 to 1093.6 ppm (comparing the timber and digested fibre). Bleaching with hydrogen peroxide was found to reduce both copper and chromium in the fibre and further reduce the arsenic content. In contrast, ultrasonic treatment had the least effect on the heavy metal content.

The variable changes in heavy metal concentration (i.e., comparing changes in Cu, Cr, and As concentration) in the digested fibre are likely a result of how the heavy metals are fixed into the timber. The literature is very unclear on the exact chemistry of the fixation of preservatives into timber; however, a recent report suggests that the heavy metals are fixed in the timber in the forms of chromium (III) arsenate, chromium (III) hydroxide and wood carboxylate–copper (II) complexes [35]. The removal of 99.8% of As suggests the removal of the majority of the chromium (III) arsenate in the timber during the digestion process. The potential reduction of chromium (III) arsenate in the timber would agree with the results; although, As is almost completely removed, and the concentration of Cr is roughly halved, which is likely due to the chromium (III) hydroxide remaining in the fibre. Thus, the results infer that the chromium (III) arsenate is fixed primarily into the alkali soluble constituents of the timber (such as hemicellulose and lignin). However, the concentration of copper and chromium changes less, which could result from chromium (III) hydroxide and carboxylate–copper (II) complexes potentially being evenly distributed and fixed throughout all the timber constituents (including cellulose).

When considering the yields of the digestion, ultrasonic, and bleaching processes, the mass of heavy metals in the fibres compared to the original mass of heavy metals in the timber is significantly reduced. Therefore, a significant portion of the heavy metals from the timber can be expected to be in the fibre treatment solutions after processing

(digestion, bleaching, and ultrasonic treatment solutions). The presence of the heavy metals will likely alter any potential commercial-scale recovery of water, biomass, or chemicals from the solutions, and will require further research. The highly alkaline solutions used for processing may precipitate the heavy metals allowing for their removal [36]. However, more research is required to investigate this possibility.

### 3.2. Characterisation of Composites

Composites were assessed for their mechanical properties and for the potential of leaching. Table 4 and Figure 5 show the tensile properties of the composites. Figure 5 shows a typical stress–strain curve from the tensile testing of polypropylene used as a matrix material, 30 wt% DB fibre composites and 40 wt% DB fibre composites. The DB 30 wt% fibre composites is representative of the curve from all of the 30 wt% composites. The figure shows that the addition of more fibre increases the strength and stiffness of the composites. However, the failure strain is reduced with the addition of more fibre.

**Table 4.** Mechanical properties of 30 wt.% F (wood flour), D (digested fibre), DU (digested and ultrasonic-treated fibre), DB (digested and bleached fibre), and DBU (digested, bleached and ultrasonic-treated fibre) composites, and 40 wt.% DB composites.

| Composite | Tensile Strength (MPA) | Failure Strain (%) | Young's Modulus (MPA) |
|---|---|---|---|
| PP—REFERENCE | 18.49 (0.03) | N/A | 835 (2.95) |
| PP F (30 wt.%) | 24.36 (1.21) [D] | 6.31 (19.7) | 1899 (6.33) [A] |
| PP D (30 wt.%) | 26.74 (1.48) [AB] | 5.26 (11.3) | 1928 (1.50) [A] |
| PP DU (30 wt.%) | 26.88 (1.53) [A] | 5.74 (17.3) | 1880 (7.16) [A] |
| PP DB (30 wt.%) | 25.37 (2.84) [CD] | 5.28 (7.35) | 1896 (6.04) [B] |
| PP DBU (30 wt.%) | 25.69 (2.67) [BC] | 6.19 (15.1) | 1883 (7.02) [A] |
| PP DB (40 wt.%) | 27.61 (2.75) | 3.75 (12.8) | 2335 (5.79) |

The COV is presented in parenthesis (%). Means that share a letter (A–D), in the same column, suggest that the differences between the values are statistically insignificant ($p < 0.05$).

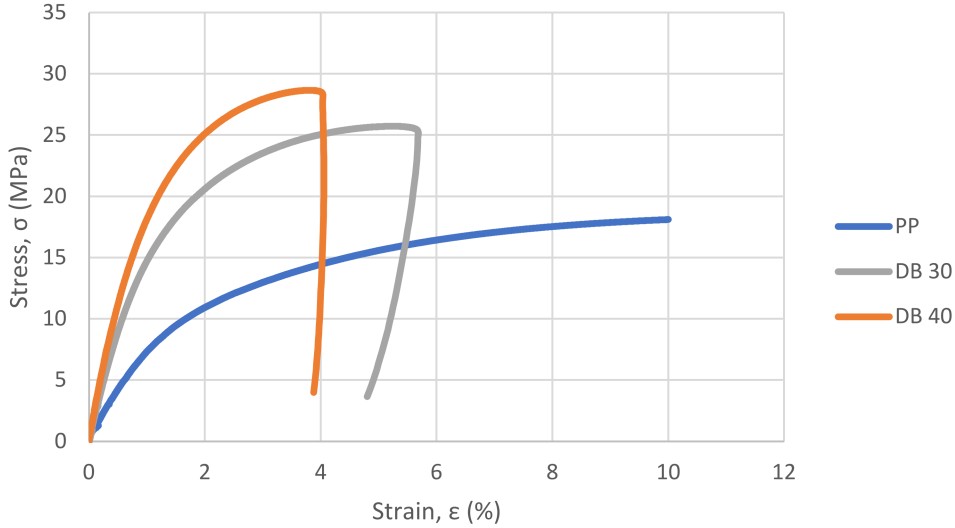

**Figure 5.** Stress–strain curve of PP 30 wt% DB fibre (DB 30) and 40 wt% DB fibre (DB 40) composites.

Table 4 shows the tensile properties of the PP and composites produced in this study. A higher tensile strength was attained in the fibre composites than wood flour composites. The highest TS obtained was from the composites produced with DU fibres, and the highest young's modulus (YM) was from the composites produced using D fibre. The results show that the bleaching process reduced the average TS and YM of corresponding composites.

When comparing composites produced with D and DB fibres, the average TS was reduced from 26.74 to 25.37 MPa, and the YM from 1928.92 to 1896.13 MPa.

The ultrasonic treatment had similar effects for both the digested and digested and bleached fibres. The inclusion of an ultrasonic treatment was found to increase the average TS and decrease the YM for composites in both cases (comparing D and DU fibre composites and DB and DBU fibre composites). The increase in TS is likely due to the increase in the surface area caused by the fibrillation from the ultrasonic treatment shown in the results of the SEM of fibre surfaces (Figure 1C,G). The increase in fibre surface area has the potential to increase interfacial strength and therefore improve composite TS. Furthermore, the reduction in YM is likely a result of the decrease in fibre length induced by the ultrasonic treatment, which was shown in the fibre quality analysis (Table 1). The reduction in length is likely to have reduced fibre reinforcement efficiency, as the length of the fibre that the load is transferred to is reduced [18]. The results from the tensile testing of the fibre composites are all very similar. The COV and the results of the Anova test indicate that the differences in the YM of the fibre composites are statistically insignificant, and the tensile strength was higher for digested fibre than digested and bleached fibre, and the use of ultraosinc treament did not make a statistically signifigant difference.

The bleached fibres had the lowest heavy metal content, likely giving an advantage as a commercially viable product. Furthermore, the TS of the composites were only slightly lower than the digested fibre composites, and there was no statistically significant difference in the YM. Composites were produced using 40 wt.% digested and bleached fibre, 4 wt.% MAPP, and 56 wt.% PP, to provide a better understanding of the mechanical performance attainable. Table 4 shows that the resulting average tensile strength was found to be 27.61 MPa, the highest strength properties attained from any composites tested during this study. The modulus increased compared to the composites with 30 wt.% bleached fibres, and a modulus of 2.33 GPa was attained. Due to the high YM and failure strain attained from the extruded 40 wt.% DB fibre composites, the composites could be regarded as having the most useable mechanical properties.

The results from the composites with 40 wt.% fibre suggest that the wetting of the fibres using the extruder is still good with the increased fibre content, therefore providing good interfacial strength and increased mechanical performance.

The failure surfaces of the composites were studied using SEM to understand the failure mechanisms during tensile testing. Overall, the composites showed minimal differences between the composites produced with varying fibre treatments. Figure 6 shows some examples of failure surfaces of the composites. The long matrix 'tongues' shown in the figures indicate that the composites fail in a ductile manner caused by plastic deformation of the matrix of the same magnitude (if not bigger) of the fibres within the composite. The composites also indicate a non-homogenous failure surface in all composites, which could potentially suggest poor mixing during the extrusion or poor compatibility with the fibres and matrix resulting in a non-homogenous mix of fibres and polymer through the composites cross-section. The figure also indicates some porosity within the composites. This could further lead to reduced composite strength, as it acts as stress concentrations in the material while testing. Overall, the SEM indicates that the composite performance could be improved with improved processing.

Creep testing was undertaken over a 12 h period to understand the influence of the introduction of fibres to PP on the viscoelastic behaviour. The 40 wt.% DB fibre composites were used for the study since they showed the most appropriate mechanical properties for the construction sector. PP was used as a comparison. The composite and PP were tested at 15 and 30% of their specific flexural strength. The PP flexural strength was 23.56 MPa, and the DB composites flexural strength was 45.19 MPa. The resultant flexural strain was measured over a 12 h period and shown in Figure 7.

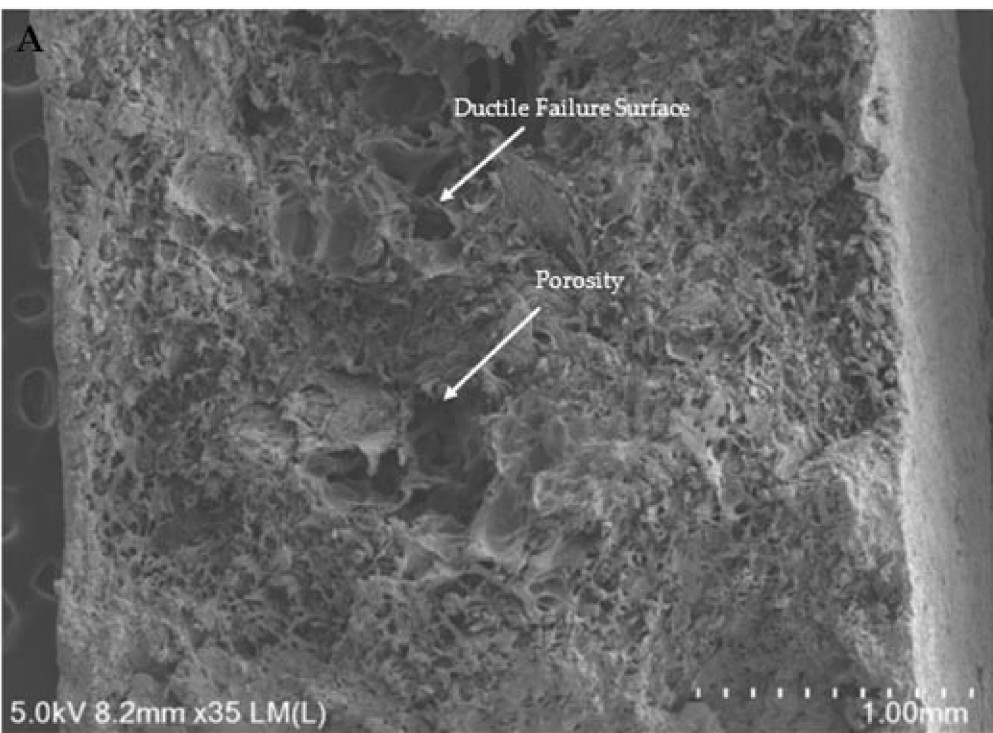

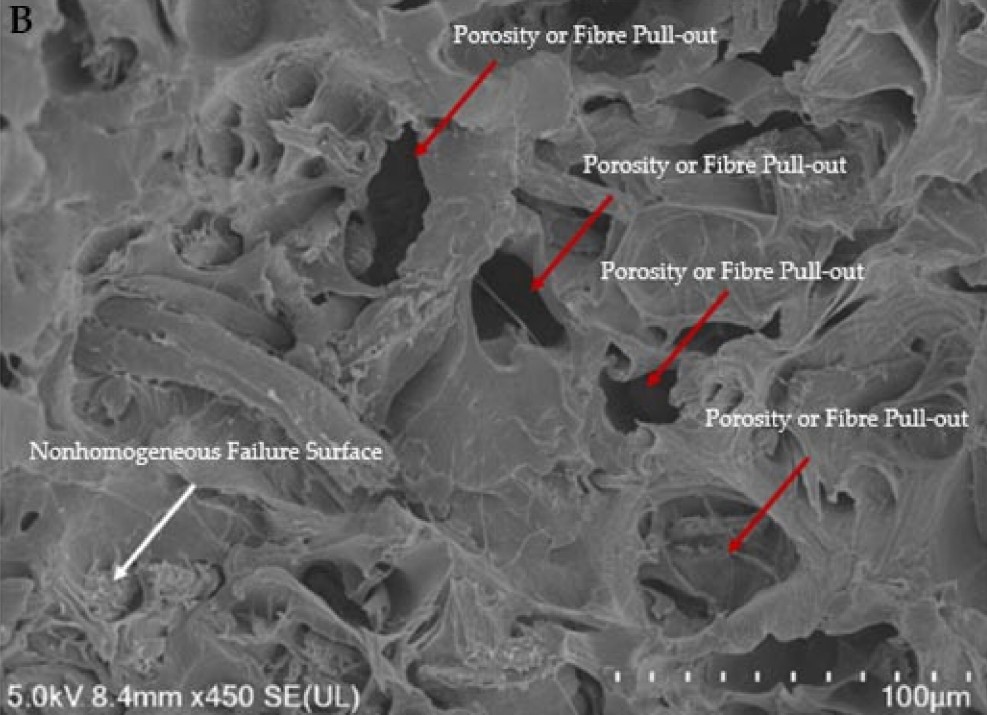

**Figure 6.** SEM of failure surfaces for (**A**) digested fibre composite with 30 wt% fibre and (**B**) digested bleached and ultrasonic fibre composites with 30 wt.% fibre.

Figure 7 shows that for tests at both 15 and 30% of the flexural strength, PP has a higher flexural strain than the composites, even though the applied stress was lower. Furthermore, the results indicate that the creep rate is higher in the PP than the DB fibre composites in both tests. Both the differences in strain and creep rate between PP and composites appear to be more pronounced in the 30% strength tests than the 15% tests.

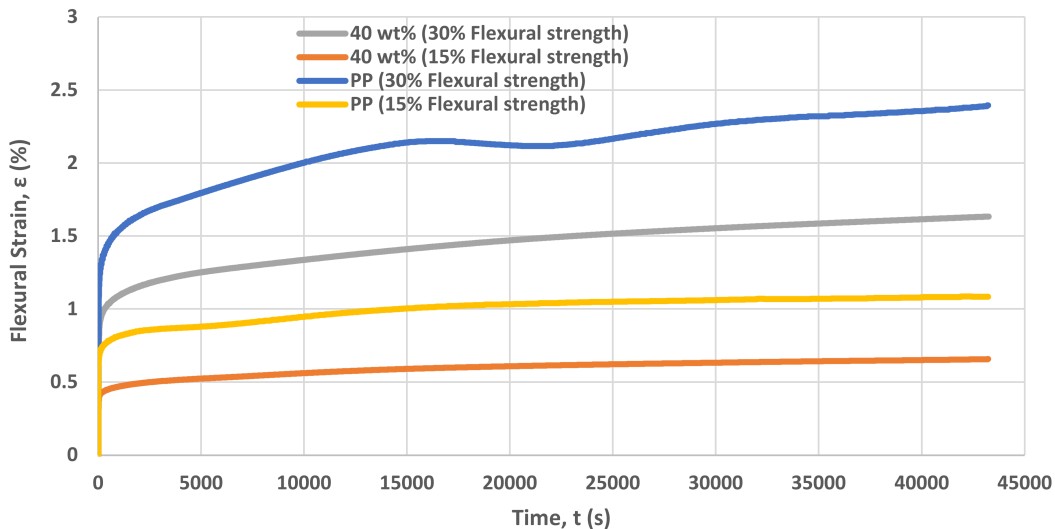

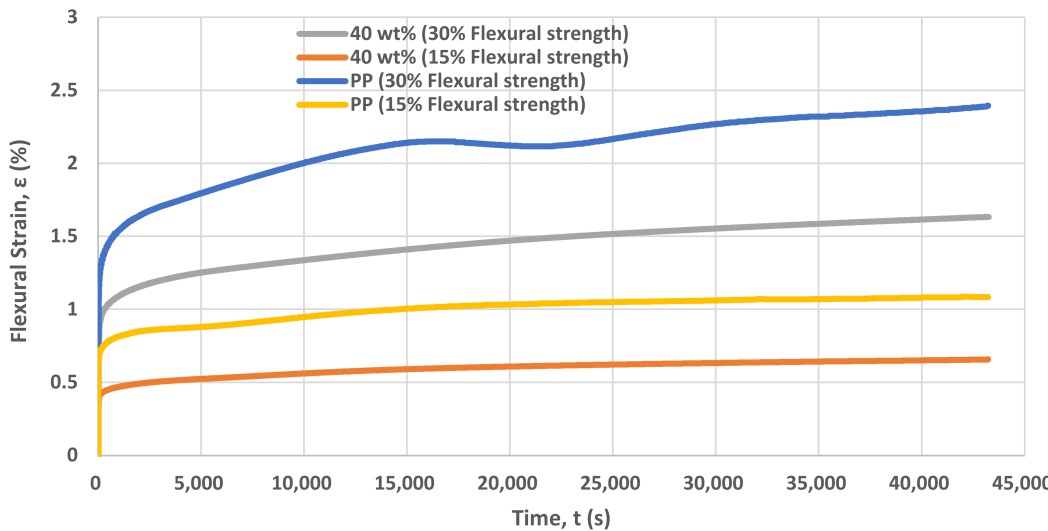

**Figure 7.** Creep testing of polypropylene (PP) and 40 wt.% DB (digested and bleached) fibre composites.

Figure 7 also shows that the creep curve for the DB fibre composites follows a smooth line that is potentially starting to plateau in both 15 and 30% tests over a 12 h period. However, the PP curve for the 30% test has an initial plateau at roughly 15,000 s before starting to accelerate again later in the test. This curvature is potentially indicative of the PP sample reaching a tertiary creep phase as the creep rate starts to accelerate at the end of the test. However, it is also possible that it is due to other factors, such as the complex semi-crystalline structure of PP. The data suggest that using the DB fibres is useful for reducing the strain in the composites and the creep rate in the material. Furthermore, it is possible that the introduction of wood fibres has increased the time taken to induce tertiary stress, which is important for structural application in the building sector, where tertiary creep should be avoided [37,38].

Variances in temperatures may have affected the test results due to the susceptibility of polymer properties to variance in temperature. However, it is unlikely that the small differences in the temperature during testing affected the trends found in the results.

The testing gives insight into the improvement of creep properties when fibres are added to polymers to alter properties. However, to understand long-term application potential, more testing and further research is required to understand creep performance in this time frame and whether tertiary creep can be avoided [39].

Figure 8 shows the cumulative leaching of copper, chromium, and arsenic from 30 wt.% wood flour (F), digested fibre (D), and digested and bleached fibre (DB) composites over a 14-day period (cumulative leaching values after 14 days are summarised in Table 5). The experimentation took place by measuring only one sample of leachate per composite material at each time interval, and the accuracy of the results could be increased through testing replicates. Furthermore, although the results of all of the testing were above the quantitative limit of the ICP–MS experimentation, some of the results indicate extremely low levels of heavy metals (especially the leaching of As from D and DB fibre composites), and the results of samples with very low levels of leaching are potentially susceptible to errors in the methods undertaken or contamination. However, the results shown in the figures give a clear trend in leaching properties between samples.

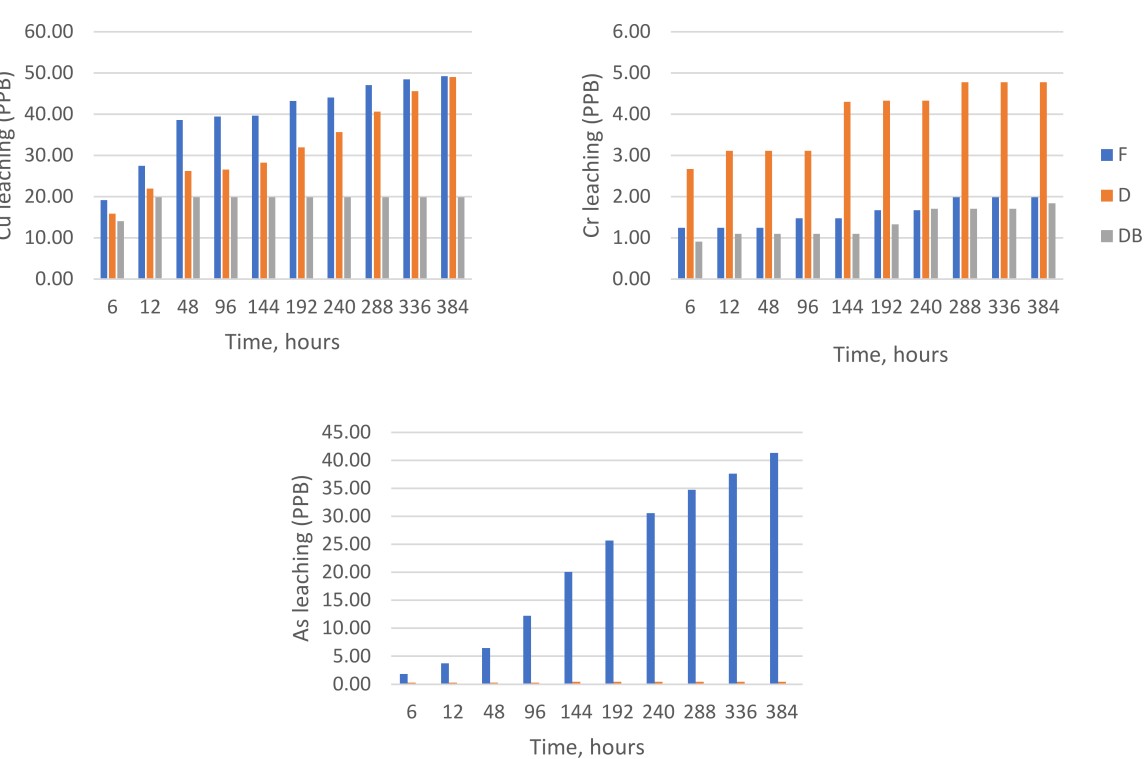

**Figure 8.** Cumulative leaching of copper, chromium, and arsenic, from 30 wt.% F (wood flour), D (digested fibre), and DB (digested and bleached fibre) extruded composites.

**Table 5.** Summary of cumulative leaching (total over 14 days) of copper, chromium, and arsenic, from 30 wt.% F (wood flour), D (digested fibre), and DB (digested and bleached fibre) extruded composites.

|  | Cu (ppb) | Cr (ppb) | As (ppb) |
|---|---|---|---|
| F | 49.22 | 1.98 | 41.29 |
| D | 48.96 | 4.78 | 0.42 |
| DB | 19.84 | 1.84 | 0.07 |

The recorded leaching values of the wood flour composites were lower than other studies, likely due to the low content of wood flour in the composites allowing for better encapsulation in the matrix, therefore restricting water penetrating the composites and reducing the ability of the heavy metals to leach out [9,10].

The results shown in Figure 8 and Table 5 show that the leaching of copper is higher from all materials than the leaching of chromium and arsenic. This is interesting as the content of chromium in both the wood flour and digested fibres is higher than the other

heavy metals, as shown in Table 3. The arsenic leaching from the wood flour was also significantly higher than the chromium leaching.

The leaching of copper and arsenic appears to be directly related to the heavy metal content in the flour and fibres. The concentration and leaching of copper were very similar for wood flour and digested fibres, and the concentration and leaching of copper were much lower for the digested and bleached fibres. The properties were very similar for arsenic; the wood flour has a high concentration of arsenic and leaches the most, whereas the D and DB fibres have very low concentrations of arsenic, and the corresponding leaching was also minimal.

Differing from the copper and arsenic leaching properties, the leaching of chromium does not appear to directly relate to the content in the wood flour and fibres. The wood flour has the highest chromium content, whereas the digested fibres appear to leach the most over a two-week period. The digested fibre composites were seen to absorb the most moisture, which has the potential to allow for more chromium to leach out of the composites. However, the leaching of chromium from the DB fibre composites suggests that there was further reasoning. Leaching and moisture absorption of the wood flour composites and the digested and bleached fibre composites appeared to be very similar; however, the content of chromium in the DB fibres is approximately five times less than that of the wood flour (353.16 ppm compared to 1728.5 ppm). Therefore, it is likely that the fixation of chromium into the composites was altered with the digestion process due to the high temperature, and pressure in the alkali solution. This could be a result of residual chromium (III) arsenate, which is expected to have been primarily removed, now being more susceptible to leaching, or an alteration in the fixation of chromium (III) hydroxide. However, it should be noted that the leaching was all measured to be in the same magnitude of ppb, and it is possible that small amounts of contamination could have caused the differences in results.

The maximum acceptable values (MAV) for the heavy metals in New Zealand's drinking water are 2000, 50, and 10 ppb for Cu, Cr, and As, respectively [40]. The leaching from composites produced with D and DB fibres both meet these standards, as seen in Table 5. Therefore, even if there were small amounts of contamination or error in the measurements, the samples are still below the MAV for safe drinking water.

The literature on using WPC produced with waste CCA-treated timber indicated that one of the largest concerns was the heavy metal content and the potential for leaching when in application. Furthermore, the two studies that investigated leaching from WPC showed leachate levels that were significantly higher than the MAV for safe drinking water and used this as a justification for their concerns. However, the results of this study suggest that the use of chemical processing to obtain wood fibres from CCA-treated timber has the potential to be used in the wood–plastic composite materials with leachate levels significantly lower than the MAV of drinking water. Therefore, the leaching test results indicate the potential for safe usage and application of composites produced with wood fibres from waste CCA-treated timber, even when in contact with people.

## 4. Conclusions and Future Recommendations

This research has evaluated the potential for using waste CCA-treated timber as a raw material for the reinforcement in WPC. The results show that the addition of fibres to PP has the potential to considerably improve composite strength, compared to PP and wood-flour-reinforced PP. The results also show that waste CCA-treated timber could be recycled to produce WPC that could be regarded as safe due to the low heavy metal content and low leaching. Additional fibre treatment was found to have minimal effect on composite mechanical performance; however, the use of hydrogen peroxide bleaching substantially decreased the heavy metal content and leaching of corresponding fibres and composites.

The use of WPC in construction has potential to reduce reliance on virgin materials (including treated timber) in the construction sector, which could further help reduce treated timber waste over time. Previous studies have indicated the potential to reprocess fibrous WPC up to eight times with minimal loss in mechanical performance, thus indicating

potential to recycle the WPC and keep it in the construction sector for long periods of time in the circular life cycle [7].

Further research on the durability of the WPC produced from waste treated timber should be undertaken, as it is likely to influence application potential. Further research would be required to investigate the recovery or disposal of heavy metals, chemicals, and water from this procedure to understand the economic and environmental implications of the processing and the potential of WPC produced from waste CCA-treated timber as a sustainable and commercially viable material.

Life cycle analysis (LCA) should also be undertaken to investigate the environmental impacts of producing WPC from waste CCA-treated timber in comparison to the use of virgin materials in the construction sector to understand the full environmental impact, including carbon emissions and ecotoxicity. LCA should be completed with a comparison of the full life cycle of traditionally used materials in construction with the complete lifecycle of WPC used in the construction sector including all potential recycling stages.

**Author Contributions:** Methodology, J.N.; investigation, J.N.; Data curation, J.N.; Writing-original draft, J.N.; writing—review and editing, K.L.P. and M.D.H.B.; supervision, K.L.P. and M.D.H.B.; validation, M.D.H.B. All authors have read and agreed to the published version of the manuscript.

**Funding:** Project was funded from the Ministry of Business, Innovation and Employment (MBIE) New Zealand Endeavour Fund, through the project Āmiomio Aotearoa—A circular Economy for the wellbeing of New Zealand (UOWX2004).

**Data Availability Statement:** Further information can be found at: https://researchcommons.waikato.ac.nz/handle/10289/15030 (accessed on 19 January 2023).

**Acknowledgments:** The authors would like to acknowledge Christian Gauss, Akindoyo John, Joanathan van Harselaar and Helen Turner for their technical support.

**Conflicts of Interest:** The authors declare no conflict of interest.

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
