# Peer review of "Assessment of the Potential of Waste Copper Chromium and Arsenic (CCA)-Treated Timber Fibre Reinforced Polypropylene Composites for Construction"

_jcs, doi:10.3390/jcs7020048_

Round 1

Reviewer 1 Report

Very interesting work. Some revisions:

1.In page 5, the authors should add a comment about IF there has been followed a standard about the conditioning of the samples.

2.The authors should report in base of what standard was conducted the flexural mechanical test and the corresponding specimen geometry.

3.The tensile stress-strain curves obtained should be presented and commented

4. The results in Fig.8 should be more presentable, not directly from Excel without elaboration.

Author Response

Very interesting work. Some revisions:

Thanks for the compliment and suggestions.

1.In page 5, the authors should add a comment about IF there has been followed a standard about the conditioning of the samples.

The condition of composites before testing was completed based on common procedures. The conditions of conditioning have been elaborated on in page 4 line 198 – 200.

2.The authors should report in base of what standard was conducted the flexural mechanical test and the corresponding specimen geometry.

The standards and dimensions of specimens have been added to page 5 line 213 – 215.

3.The tensile stress-strain curves obtained should be presented and commented

The tensile stress-strain graph has been added in page 15 as well as a brief comment in line 486 – 492. All following Figure numbers have been updated.

4. The results in Fig.8 should be more presentable, not directly from Excel without elaboration.

The data has been changed to 2 d.p. or 4 s.f.

Reviewer 2 Report

In my opinion, this research paper contains significant and relevant information that justify publication, with minor revision regarding the following issues:

1.     Abstract:

Page 1 –line 10-11: missing acronym: “...copper chromium and arsenic - treated timber...” should be ““...copper chromium and arsenic (CCA) treated timber...”, as in following line 18 the authors used the acronym.

2.     Introduction

The content is succinctly described and contextualized with respect to previous and present theoretical background on the topic and supported by relevant references on the topic,

Page 1 –line 38: missing acronym: “...raw material for WPC...” should be ““...raw material for wood-plastic composite (WPC)...”, first time used in text, maintaining the logic of former presentation of acronym presentation (in line 30).

Page 2-line 61:  “The use of MAPP as a coupling agent ...” should be “ The use of Propylene Maleic Anhydride Copolymer (MAPP)...”

Page 2 – line 85: “... kenaf fibre and PLA composite ...” should be “... kenaf fibre and Poly(lactic acid) (PLA) ...”

3.     Materials and Methods:

1)       It is not explicit how many replicates were performed in each analysis, only in page 5 line 203- 204 was referred that “   A minimum  of four specimens were tested per condition.” In 2.2.4 section

2)        a statistical analysis section is missing in order to properly explain the statistical methodology used .

4.     Results and Discussion

1)       Page 7: Figure 1 I should be revised, annotation in Figure I not readable (formatting )

2)       Page 7  from line 284- 292 – Discussion can be should be reinforced by statistical analysis of data. Suggestion : revised discussion, evidencing statistical analysis

3)       Page 9 – Table1 – please revise Table formatting. , revised unit of mean width (missing Symbol letter style (“um” instead of “mm”)

4)       Page 9 line 336 – “Pinus Radiata” should be written as “Pinus radiata” (plant species scientific notation)

5)       .Page 10 – line 347 – “... NaOH gives no significant change   “ – without statistical analysis evidenced, one cannot say that is significant. Or present (p<0,05, 0,001, confidence level)

6)       Page 13 – Table 2 - please revise Table formatting. Data presented are mean data?, if so, why not present standard deviation or standard error?

7)       Page 14 – 438: “... was found to significantly reduce ...” - without statistical analysis evidenced, one cannot say that is significant. Or present (p<0,05, 0,001, confidence level)

8)       Page 14 – line 466-468  Table 3 legend should include if results presented are mean values (revised legend, the same in Table 1 and 2)

9)       Page 15 – Table 4 – please revise table 4 – information regarding statistical analysis is missing (letters D and E are missing). Information “Means that share a letter (A, B, C, D, E, or F), suggest that he differences are statistically insignificant.” Should be “Means that share a letter (A, B, C, D, E, or F), in the same column, suggest that the differences are statistically insignificant (p< 0,05).” (confidence level should be presented

10)    Page 6 – line495-496 : “...statistically  insignificante.” Should be “statistically  insignificante (p>0,05).” (include confidence level, it should be clear if 0,05 or 0,01 or 0,001)

11)    Page 20 line 585 – formatting problem, text ended in uncompleted sentence in  the the middle of the page. Please revise

12)    Page 20 – Table 5 - please revise Table formatting. Data presented are mean data?, if so, why not present standard deviation or standard error?

5.     References

Please revise reference 1, 4, 5, 6, 28 and 39 – not in accordance with the journal guides for the authors

Author Response

In my opinion, this research paper contains significant and relevant information that justify publication, with minor revision regarding the following issues:

  1. Abstract:

Page 1 –line 10-11: missing acronym: “...copper chromium and arsenic - treated timber...” should be ““...copper chromium and arsenic (CCA) treated timber...”, as in following line 18 the authors used the acronym.

Completed

  1. Introduction

The content is succinctly described and contextualized with respect to previous and present theoretical background on the topic and supported by relevant references on the topic,

Page 1 –line 38: missing acronym: “...raw material for WPC...” should be ““...raw material for wood-plastic composite (WPC)...”, first time used in text, maintaining the logic of former presentation of acronym presentation (in line 30).

Page 2-line 61:  “The use of MAPP as a coupling agent ...” should be “ The use of Propylene Maleic Anhydride Copolymer (MAPP)...”

Page 2 – line 85: “... kenaf fibre and PLA composite ...” should be “... kenaf fibre and Poly(lactic acid) (PLA) ...”

The above names and acronyms have been fixed based on the reviewers comments.

  1. Materials and Methods:

1)       It is not explicit how many replicates were performed in each analysis, only in page 5 line 203- 204 was referred that “   A minimum  of four specimens were tested per condition.” In 2.2.4 section

Replicate information has been added or changed for ICP-MS pg 4 line 189-190,tensile testing (pg 5 line 207-208), creep testing (pg 5 line 218-219), and leaching (pg 5 line 232-233).

2)        a statistical analysis section is missing in order to properly explain the statistical methodology used .

A statistical analysis section is added in page 5 line 234 – 241.

  1. Results and Discussion

1)       Page 7: Figure 1 I should be revised, annotation in Figure I not readable (formatting )

The Figure has been formatted so the annotations can be read.

2)       Page 7  from line 284- 292 – Discussion can be should be reinforced by statistical analysis of data. Suggestion : revised discussion, evidencing statistical analysis

The discussion is about the figures and no data (or statistical analysis) has been presented here yet.

3)       Page 9 – Table1 – please revise Table formatting. , revised unit of mean width (missing Symbol letter style (“um” instead of “mm”)

The Fibre width has been updated to show µm

4)       Page 9 line 336 – “Pinus Radiata” should be written as “Pinus radiata” (plant species scientific notation)

The formatting has been updated based on the reviewers suggestion.

5)       .Page 10 – line 347 – “... NaOH gives no significant change   “ – without statistical analysis evidenced, one cannot say that is significant. Or present (p<0,05, 0,001, confidence level)

The sentence has been updated to “the figure suggests … to give no significant change” a it I based on qualitative analysis without statistics.

6)       Page 13 – Table 2 - please revise Table formatting. Data presented are mean data?, if so, why not present standard deviation or standard error?

The data provided in Table 2 is based on measurements taken from one TGA sample, based on common practice in TGA analysis. (no change has been made)

7)       Page 14 – 438: “... was found to significantly reduce ...” - without statistical analysis evidenced, one cannot say that is significant. Or present (p<0,05, 0,001, confidence level)

Changed to “found to reduce”

8)       Page 14 – line 466-468  Table 3 legend should include if results presented are mean values (revised legend, the same in Table 1 and 2)

Table caption now includes “mean”

9)       Page 15 – Table 4 – please revise table 4 – information regarding statistical analysis is missing (letters D and E are missing). Information “Means that share a letter (A, B, C, D, E, or F), suggest that he differences are statistically insignificant.” Should be “Means that share a letter (A, B, C, D, E, or F), in the same column, suggest that the differences are statistically insignificant (p< 0,05).” (confidence level should be presented

10)    Page 6 – line495-496 : “...statistically  insignificante.” Should be “statistically  insignificante (p>0,05).” (include confidence level, it should be clear if 0,05 or 0,01 or 0,001)

Answering 9 and 10. The Anova test was originally run for a greater set of data thus requiring more letters. The statistical test has been reproduced for the set of data and presented in Table 4. Due to slight changes in the results caused by the smaller set of data (less comparisons) the document has been reivsed in lines 519 – 522 and 524 – 526.

11)    Page 20 line 585 – formatting problem, text ended in uncompleted sentence in  the the middle of the page. Please revise

Fixed.

12)    Page 20 – Table 5 - please revise Table formatting. Data presented are mean data?, if so, why not present standard deviation or standard error?

The data is not mean data. It is the only data taken as described in lines 590 – 592.

  1. References

Please revise reference 1, 4, 5, 6, 28 and 39 – not in accordance with the journal guides for the authors

The references have been revised following the journal guides

Round 2

Reviewer 1 Report

The authors corrected the points of the paper indicated by the reviewer. They should also correct the Fig.5 (stress-strain curves) in which the axis scales start from minus values (they should both start from 0). This way the graph will be more presentable and not a copy-paste from Excel.

Author Response

Figure 5 has been edited so both axis start at zero.

Reviewer 2 Report

The revised paper is in conditions to be published. The authors have performed the revision formerly suggested.

Only a very few additional corrections to be performed:

Page 16 – line 500,  Table 4 – “Means that share a letter (A, B, C, D, E, or F), suggest that he differences are statistically insignificant.” Should be “Means that share a letter (A, B, C, D, E, or F), in the same column, suggest that the differences are statistically significant (p< 0.05).”

(Due to my fault, the text suggested to the authors contained an error, please accept my apologies)

 Page 21 line 611 – Formatting problem, text ended in uncompleted sentence: The results shown in Figure 8 and     “ This sentence continues in page 22- line 612. Please revise

Author Response

Page 16 – line 500,  Table 4 – “Means that share a letter (A, B, C, D, E, or F), suggest that he differences are statistically insignificant.” Should be “Means that share a letter (A, B, C, D, E, or F), in the same column, suggest that the differences are statistically significant (p< 0.05).”

(Due to my fault, the text suggested to the authors contained an error, please accept my apologies)

Values sharing a letter are statistically insignificant. We have reviewed the sentence to make this clearer. 

 Page 21 line 611 – Formatting problem, text ended in uncompleted sentence: The results shown in Figure 8 and     “ This sentence continues in page 22- line 612. Please revise

This does not appear on the manuscript on my computer.

Round 3

Reviewer 1 Report

The reviewer is satisfied with the minor changes done by the authors. The work may be published in the current form.

Reviewer 2 Report

The revised paper is in conditions to be published. The authors have performed the revision formerly suggested. No further comments